# Beyond Harmful: Exploring Biofilm Formation by Enterococci Isolated from Portuguese Traditional Cheeses

**DOI:** 10.3390/foods13193067

**Published:** 2024-09-26

**Authors:** Susana Serrano, Maria Vitória Ferreira, Cinthia Alves-Barroco, Susana Morais, Maria Teresa Barreto-Crespo, Rogério Tenreiro, Teresa Semedo-Lemsaddek

**Affiliations:** 1CIISA—Center for Interdisciplinary Research in Animal Health, Faculty of Veterinary Medicine, University of Lisbon, 1300-477 Lisbon, Portugal; sserrano@fmv.ulisboa.pt (S.S.); smorais@fmv.ulisboa.pt (S.M.); 2Associate Laboratory for Animal and Veterinary Sciences (AL4AnimalS), 5000-801 Vila Real, Portugal; 3Faculty of Sciences (FCUL), University of Lisbon, 1749-016 Lisbon, Portugal; 4iBET, Institute of Experimental Biology and Technology, P.O. Box 12, 2781-901 Oeiras, Portugal; tcrespo@ibet.pt; 5ITQB, Institute of Chemical and Biological Technology António Xavier, Nova University of Lisbon, Republic Avenue, 2780-157 Oeiras, Portugal; 6BioISI—Biosystems & Integrative Sciences Institute, Faculty of Sciences, University of Lisbon, 1749-016 Lisbon, Portugal; rptenreiro@ciencias.ulisboa.pt

**Keywords:** traditional cheese, autochthonous microbiota, *Enterococcus* spp., biofilm formation

## Abstract

This study investigated the biofilm-forming capabilities of *Enterococcus* isolates from Portuguese traditional cheeses with protected designation of origin (PDO) status, specifically *Azeitão* and *Nisa*. Given the absence of added starter cultures in the cheesemaking process, the characteristics of these cheeses are intrinsically linked to the autochthonous microbiota present in the raw materials and the production environment. Our findings demonstrate that all isolates possess biofilm production abilities, which are crucial for their colonization and persistence within cheese factories, thereby maintaining factory-specific microbial heritage. Through an integrated analysis utilizing principal component analysis (PCA), a direct correlation between biofilm formation and cell viability was established. Notably, these results underscore the adaptive capacity of enterococci to survive environmental fluctuations and their role in the unique characteristics of Portuguese traditional cheeses. Overall, this research enhances our understanding of the microbial dynamics in cheese production and highlights the importance of enterococci in preserving cheese quality and heritage.

## 1. Introduction

Cheese is a fermented product with an ancient history and a long-standing relevance in the human diet, due to its organoleptic features and nutritional composition, associated with multiple health benefits [1,2,3,4]. Cheese production, particularly that of traditional cheese, is a complex process that involves a number of stages, including coagulation, syneresis, salting (in certain cheese varieties), and ripening [5]. The type of milk used (bovine, caprine, or ovine) and the steps and timings followed during cheese production will significantly impact the final product’s taste, color, texture, and overall composition. The associated microbiota and environmental conditions in which the product is manufactured, including the humidity, salt content, and pH, also contribute to the final product’s specific features. The ripening stage is of particular importance in this regard as it is during this period that distinctive aspects of texture and flavor fully mature [5,6,7].

In Portugal, a number of cheeses are produced in rural areas using traditional manufacturing techniques passed down from generation to generation, resulting in the development of unique products. The distinctiveness of these fermented dairies, along with their economic importance, granted the protected designation of origin (PDO) status to various traditional cheeses [6,8,9]. Briefly, PDO identifies a product that originates from a defined place or region with distinct and unique characteristics, stipulating that “every part of the production, processing and preparation process must take place in the specific region” [10].

A plethora of Portuguese PDO cheeses, such as *Azeitão* and *Nisa* cheeses, are manufactured using ewe’s raw milk, *Cynara cardunculus* L. as a coagulation agent, and salt, without the addition of starter microbiota [6,8,9], the predominant autochthonous microorganisms being lactic acid bacteria (LAB) [9,11]. LAB include a highly diverse group of microorganisms able to convert lactose into lactic acid during milk fermentation, as well as perform a series of additional biochemical reactions mainly associated with ripening stages [12], which confer specific organoleptic characteristics to the final product, while contributing to the overall safety and related health benefits [13,14,15,16]. LAB present in dairy products include bacteria from various genera, such as *Lactococcus*, *Lactobacillus*, *Leuconostoc*, and *Enterococcus* [9,17]. Among cheese autochthonous microbiota, *Enterococcus* spp. are undoubtedly the most controversial. In contrast with other LAB, enterococci do not harbor the qualified presumption of safety (QPS) status, attributed by the European Food Safety Authority (EFSA), being the subject of extensive studies with the objective of determining whether specific strains may be safe for use in foods [18,19,20].

*Enterococcus* spp. are ubiquitarian in a variety of settings, including water, soil, and the gastrointestinal tract of both humans and animals [21,22]. Given their high adaptability to novel environments, resistance to stress conditions, and propensity for horizontal gene transfer, these bacteria pose a putative concern for human health. However, enterococci also offer a range of technological and probiotic applications, with increasing relevance in the food industry [21,23,24]. In cheese production, enterococci play a significant role during the ripening process, exhibiting proteolytic and lipolytic activities that contribute to the production of aroma compounds, which in turn influence the final product’s organoleptic features [25,26]. These microorganisms are also known for their ability to form biofilms, which is an interesting topic of research. In fact, biofilm production may be considered a hazardous trait for being associated with environmental persistence, cross-contamination, and horizontal gene transfer events, as well as equipment damage [27,28,29,30]. Nonetheless, biofilms can also have beneficial roles, such as facilitating the fermentation process in certain foods [31,32].

Biofilms can be defined as an “aggregate of microorganisms in which cells that are frequently embedded within a self-produced matrix of extracellular polymeric substance (EPS) adhere to each other and/or to a surface” [33]. The formation of biofilms is a complex process that occurs in five distinct phases: initial reversible attachment (i), irreversible attachment (ii), production of EPS (iii), maturation (iv), and dispersion (v). These structures can be monomicrobial or polymicrobial, comprising microorganisms of the same or different species [34,35]. Biofilm formation typically occurs in response to external stresses such as nutrient availability, temperature fluctuations, pH changes, and various biotic and abiotic factors. This process enhances resistance to environmental stress, as well as to antimicrobial, chemical, and sanitizer treatments [14,29,36]. This complex organization grants microorganisms increased resistance and protection, enabling them to thrive in diverse environments [33,36]. Moreover, during the dispersion phase, cells revert to a planktonic state, allowing them to colonize new surfaces and promote biofilm spread [29].

As aforementioned, in the food industry, biofilm formation is typically viewed as a detrimental phenomenon. This is because biofilms formed by, or including, potential pathogens or food-spoiling microorganisms can pose significant challenges to food safety and quality [27,28,37]. Moreover, in the food industry, the primary source of the final product contamination has been identified as biofilms formed on contact surfaces [31,38,39]. However, recent studies have indicated that biofilm formation can also have advantageous roles, such as enhancing plant health and productivity by acting against phytopathogens or serving as biofertilizers [35,40]. Additionally, biofilms facilitate wastewater treatment and offer various biotechnological applications, including the production of edible biofilms, their use as probiotics, and the control of biofilm formation by pathogenic bacteria [35,37,40,41,42,43].

Previous studies [4,20,43,44,45,46,47] have demonstrated the ability of LAB from milk and cheese, particularly *Enterococcus* spp. and *Lactobacillus* spp., to form biofilms. Focusing on enterococci, these bacteria have been identified in biofilms within the food industry due to their ability to adhere to a variety of surfaces [4,48]. Moreover, during the manufacturing of traditional cheeses, it is common practice for cheeses from different batches to be aged on the same shelves in the maturation chamber without rigorous abrasive cleaning. This practice allows microbial biofilms, composed of microorganisms native to the cheesemaking environment (autochthonous microbiota), to persist over time. Reports from various traditional cheese producers (personal communications) and studies by Galié et al. [49] indicate that these autochthonous microorganisms are crucial for developing the distinctive features of these cheeses and maintaining their organoleptic properties over time.

To further explore this topic, the present study aims to investigate the biofilm production capabilities of *Enterococcus* isolates from *Nisa* and *Azeitão* PDO cheeses under various incubation conditions relevant to cheese production.

## 2. Materials and Methods

### 2.1. Microbial Collection

The microbial collection used in this study was composed by cheese enterococci isolated from Portuguese PDO cheeses of *Azeitão* (38.5194° N, 9.0138° W) and *Nisa* (39.5180° N, 7.6484° W), produced from 2016 to 2022 (excluding the year 2020 due to COVID-19 pandemic contingencies). Note that isolation details have been previously described by Rocha et al. [23].

For preliminary analysis, a total of 865 cheese enterococci were submitted to RAPD-PCR, as described earlier [23], leading to the selection of 72 genetically distinct enterococci, representing all sampling years, regions, and producers. Subsequently, genus and species allocation were performed as previously described [50,51].

The details of the enterococcal collection analyzed in the present study are depicted in Appendix A, including 53 isolates from *Azeitão* and 19 from *Nisa*. It is important to note that for cheese factory A5, PDO cheese production occurred exclusively in 2016 and 2017, so only isolates from this period were included in the study. Furthermore, the enterococci recovered in 2021 and 2022 exhibited greater diversity compared with earlier years, leading to the selection of various isolates.

### 2.2. Biofilm Formation

The ability to form biofilm was assessed according to Extermina et al. [52], with a few modifications, described below. As the control, bacteria from international culture collections (CECT (Colección Española de Cultivos Tipo) and ATCC (American Type Culture Collection)), such as *Listeria monocytogenes* CECT 934 and *E. faecalis* ATCC 700802, were used. All assays were performed in three independent days (biological replicates), and each microplate included three repetition wells (technical replicates).

#### 2.2.1. Bacterial Suspension

Bacterial suspensions were prepared to a final concentration of 1 × 10^9^ CFU/mL. Briefly, each isolate was grown in 5 mL of Brain Heart Infusion (BHI, Scharlau) broth for 24 h at 37 °C and centrifuged at 13,400 rpm for 3 min at 4 °C (Z233 M-2, HermLe), the supernatant was discarded, and the pellet was resuspended in 1 mL of Ringer’s solution (Liofilchem). As a washing step, the suspension was resuspended and subsequently centrifuged again for 5 min at 13,400 rpm. After centrifugation, the supernatant was discarded, and the pellet was resuspended in 500 µL of Ringer’s solution. The same protocol was performed for each condition tested throughout the present research study.

#### 2.2.2. Biofilm Assays

For the evaluation of biofilm production, 96-well polystyrene microtiter plates (Nunc™, Thermo Fisher Scientific Inc., Roskilde, Denmark) were used. Each well was filled with 180 µL of Tryptic Soy broth (TSB, Biokar diagnostics, Braeside, Australia) supplemented with 1.5% glucose (Biochem, Chemopharma, Burgundy, France). Subsequently, 20 µL of each bacterial suspension was used to inoculate the microtiter plates, including biological and technical replicate wells. For sterility control, only culture media was used. After inoculation, the plates were incubated at 37 °C for 24 or 48 h. The same protocol was repeated for the remaining conditions tested in this study, temperatures (4, 10, 20, and 37 °C), NaCl concentrations (1, 2, 4, and 8%), and pH values (5, 6, 7, 8, and 9).

For biofilm quantification, two distinct complementary approaches were employed: cell viability assays using 0.01 µg/mL of resazurin (RZ, Scharlau, Barcelona, Spain) and biofilm quantification using 0.1% (*w*/*v*) crystal violet (CV, Merck) staining, following the procedures and protocols detailed below.

##### Cell Viability

Viability assessment was performed as described by Aksoy et al. [53], with some modifications. After the incubation period (24 or 48 h), the growth media of each plate was discarded by inversion, followed by two washing steps with Ringer solution [30,37,54,55,56]. Subsequently, the biofilm was fixated by heat at 50 °C for 30 min [56,57,58]. After fixation, RZ at 0.01 µg/mL (prepared in Ringer solution) was added to each well, and the absorbance at an optical density (OD) of 570 nm and 600 nm was assessed in a microplate reader (FLUOstar OPTIMA, BMG LABTECH). Thereafter, the plates were incubated at 37 °C for 150 min [53], followed by a final absorbance read at OD_570_ and OD_600_ [59]. Additionally, during this incubation period, the microplates were observed for any visual alterations; if the bacteria were metabolically active, resazurin, which is blue in color, would be reduced to resofurin, resulting in a color change to pink [60,61].

To calculate the percentage of viable cells in the biofilms, fluorescence reading is the recommended method. However, due to equipment unavailability, only absorbance readings were taken. Consequently, as described by Goegan et al. [62], a correction of the absorbances was applied. Briefly, the reduced resazurin (AR570) was calculated using the formula below. The ratio of absorbance at 570 nm to the absorbance at 600 nm for the oxidized substrate (Ro) was used as a correction factor [62,63].
$${\mathrm{AR}}_{570}={\mathrm{OD}}_{570}-\left({\mathrm{OD}}_{600}\times \mathrm{Ro}\right)\times 100$$

##### Biofilm Quantification

After the viability assessment, the microplates were washed twice by submersion in distilled water and then dried at room temperature for 15 min. Next, 100 µL of 0.1% crystal violet (prepared in sterilized distilled water) was added to each well and incubated at room temperature for 15 min. Following incubation, the microplates were washed twice with distilled water and dried again at room temperature for 15 min. To solubilize the biofilm, a mixture of ethanol and acetone (80:20) was added, followed by a 15 min incubation at room temperature. Finally, the OD_570_ was measured using a microplate reader (FLUOstar OPTIMA, BMG LABTECH, Cary, NC, USA).

To further classify the enterococci as strong, moderate, or weak biofilm producers, a classification system was established based on the methodologies of Mohamed et al. [64] and Stepanovic et al. [56]. For this classification, the cutoff value (ODc), which corresponded to the optical density of the sterility well for each condition tested, was determined and calculated using the formula below. After ODc calculation the following thresholds were defined, non-producer (OD < ODc), weak producer (ODc < OD < 2ODc), moderate producer (2ODc < OD < 4ODc), and strong producer (OD > 4ODc).
$${\mathrm{OD}}_{c}=\mathrm{average}\;\mathrm{OD}\;\mathrm{of}\;\mathrm{control}\;\mathrm{wells}+3\times {\;\mathrm{SD}}_{c}$$

#### 2.2.3. Data Analysis

A statistical analysis was conducted on the different biofilm conditions tested. Since the data did not follow a normal distribution, non-parametric statistical tests were employed. The data obtained from the assays conducted over various incubation periods were analyzed using the RankProd statistical test using R software version 4.2.1. Additionally, radar charts were created to represent data for both detection methods, crystal violet and resazurin, under different conditions, utilizing the fmsb package (version 0.7.6) also in R [65].

Subsequently, principal component analysis (PCA) was applied to the remaining tests/conditions to determine whether the behavior of the microorganisms was similar, or markedly different, under the various conditions tested [66,67]. PCA was performed using NTSYS software version 2.2, resulting in a three-dimensional diagram. A similarity dendrogram was also constructed, employing Euclidean distance as a measure of association and the agglomeration method based on the unweighted pair group method with arithmetic mean (UPGMA).

Ultimately, to determine biofilm absorbance units per viable cell, the biofilm quantified by crystal violet was divided by the cell viability detected by resazurin. Isolates for which no viability could be quantified were marked with an asterisk (*).

## 3. Results and Discussion

### 3.1. Biofilm Formation: Preliminary Assays

The enterococcal collection was assessed for its ability to produce biofilms in TSB with 1.5% glucose at 24 and 48 h, based on biofilm staining with crystal violet and subsequent absorbance measurements. The isolates were categorized as non-producers or weak, moderate, or strong biofilm producers, according to the classification described in the Biofilm Quantification Section. Overall, out of the 72 enterococci, 53 were phenotypically classified as biofilm producers (Table 1 and Figure 1) under at least one of the incubation conditions tested, with the majority identified as *E. faecalis*.

At 24 h, 58% (42/72) of the isolates were classified as non-producers, 29% (21/72) as weak producers, 10% (7/72) as moderate producers, and 3% (2/72) as strong producers. The moderate producers included A5.18.2017 (*E. faecalis*), A1.40.2016 (*E. faecalis*), A3.34.2016 (*E. faecium*), A4.19.2016 (*E. durans*), A4.26.2021 (*E. faecalis*), N10.21.2021 (*E. faecalis*), and A3.38.2022 (*E. faecalis*). The two strong producers, A1.6.2017 and A1.14.2022, belonged to the species *E. durans* and *E. faecalis*, respectively.

At 48 h, there was an increase in biofilm formation, with 32 weak and 17 moderate producers identified (Table 1). Among the two strong producers at 24 h, only A1.14.2022 exhibited strong biofilm production at 48 h, while A1.6.2017 was reclassified as a weak producer.

The statistical analysis using RankProd allowed for the identification of 16 isolates that showed significant differences (*p* < 0.05), when comparing the biofilm production ratio between 24 and 48 h, as follows: A1.40.2016, A2.48.2016, A3.34.2016, A4.19.2016, A1.6.2017, A2.17.2017, A5.18.2017, A2.20.2019, A3.2.2019, N10.14.2019, A1.24.2021, A4.26.2021, N9.58.2021, N10.21.2021, N10.45.2021, and A3.14.2022 (Appendix A).

A study by Gajewska et al. [4] also evaluated the ability of *Enterococcus* isolated from both raw milk and cheese manufactured with unpasteurized milk, to produce biofilms using crystal violet as the detection method. Briefly, the raw milk cheese microbial collection included a total of 36 enterococci: 29 *E. faecalis*, 5 *E. faecium*, and 2 *E. gallinarum*. The authors reported that half were considered non-producers. In our study the percentage of non-producers was lower; around 29% of the isolates were not able to produce biofilms after 48 h of incubation (Table 1). Moreover, the authors also reported that the remaining enterococci were classified as strong biofilm producers and that *E. faecium* isolates were the major contributors [4]. These results were not consistent with those obtained in the present study, since there were some isolates considered weak or moderate producers, and in addition, the major contributor was *E. faecalis*. Results more equivalent to ours were reported by Pereira et al. [68] where a study on *Enterococcus* isolates from buffalo’s raw milk, communicated higher biofilm production by *E. faecalis*, in comparison with *E. faecium*. Overall, these results suggested that biofilm formation may be a strain-specific trait, rather than a genus- or species-related feature.

Subsequently, to facilitate testing under a wide variety of incubation conditions, the enterococcal collection was reduced. This selection aimed to preserve the diversity of biofilm-producing abilities while ensuring the representation of different species, cheese factories, and production years. Hence, *ca.* 10% of the isolates were selected as follows (Table 2): A2.48.2016 (*E. durans*), N9.25.2016 (*E. faecium*), A1.6.2017 (*E. durans*), N10.27.2017 (*E. faecium*), A4.16.2019 (*E. faecalis*), N9.1.2019 (*E. faecalis*), A4.26.2021 (*E. faecalis*), N10.1.2021 (*E. faecalis*), A1.14.2022 (*E. faecalis*), and N10.55.2022 (*E. faecalis*). Table 2 presents the classification assigned to each isolate, following the biofilm production assay with incubation periods of 24 or 48 h at 37 °C.

### 3.2. Biofilm Formation at Different Temperatures

The selected isolates were subsequently studied for their ability to produce biofilm at different temperatures (4, 10, 20, and 37 °C) over incubation periods of 24 and 48 h. These conditions were intentionally chosen to mimic storage and maturation temperatures (4 and 10 °C) [9], as well as room and human body temperatures (20 and 37 °C) [8]. In general, it was observed that biofilm production increased with rising temperatures, particularly at the 48 h mark, although some exceptions were noted (Table 3 and Figure 2).

Overall, the obtained results identified the temperature of 37 °C as optimal for biofilm production among the majority of isolates, with a preference for a 48 h incubation period. At 24 h, isolates A2.48.2016, N9.25.2016, N10.27.2017, and N10.1.2021 were classified as non-producers. However, at 48 h, only N9.25.2016 and N10.27.2017 remained classified as non-producers, consistent with our preliminary assays (Table 2). The remaining isolates were categorized as either weak or moderate producers, with the exception of A1.14.2022, which was classified as a strong producer, aligning with the results of the previous assay (Table 1). At 20 °C, the ideal incubation period was 48 h, during which only isolates A2.48.2016, N9.25.2016, N10.27.2017, and A4.26.2021 were classified as non-producers (Table 3 and Figure 2). The remaining enterococci were categorized as weak producers. At the lower temperatures of 4 and 10 °C, most isolates were classified as non-producers, regardless of the incubation period. The exceptions were N9.25.2016 at 48 h at 4 °C and N10.1.2021 at 48 h at 10 °C, which were identified as weak producers. Conversely, A1.14.2022 was classified as a moderate producer at both temperatures and incubation periods (Table 3 and Figure 2).

In a study by Marinho et al. [69], the authors investigated the influence of environmental factors, specifically sugar content (0.75% glucose) and temperature (10, 28, 37, and 45 °C), on the biofilm formation abilities of *E. faecalis* and *E. faecium* isolated from vegetables, meat, and dairy products in Brazil. These conditions are commonly found not only in foods but also in food processing industries. At 10 °C, the authors observed that despite this genus’s known tolerance to a wide range of temperatures, only 20% of *E. faecalis* and 33% of *E. faecium* were able to produce biofilm, categorizing them as weakly adherent (~weak producers). Additionally, the study indicated that these enterococci were predominantly sourced from meat or dairy products. At 28 °C, higher percentages of weak adherence were reported, with approximately 54% for *E. faecalis* and 24% for *E. faecium*. Moderate adherence (~moderate producers) was noted in about 31% of *E. faecalis* and 24% of *E. faecium*. Interestingly, *E. faecium* exhibited better adaptation to the tested conditions, with 43% of isolates classified as strong producers compared with only 28% of *E. faecalis*. At 37 °C, similar to our findings, there were higher percentages of both moderate and strong adherence for both species. However, the authors did not report any non-adherent enterococci, while in our study, three isolates (one *E. durans* and two *E. faecium*) were classified as non-producers.

Another study by Chotinantakul et al. [70] examined biofilm production in enterococci isolated from fermented pork across various temperatures (4, 25, and 37 °C) and found that 37 °C was optimal for biofilm production in all species analyzed, including *E. faecalis*, *E. faecium*, and *E. hirae*. At this temperature, the authors reported the highest number of strong producers, with 13 *E. faecalis* and 1 *E. faecium* exhibiting robust biofilm production. These results aligned with those of the present study, in which all isolates classified as strong or moderate producers at 37 °C were *E. faecalis*. At 4 °C, as also assessed in our study, the authors reported no strong producers. However, they identified 3 *E. faecalis* and 2 *E. faecium* as moderate producers, along with 10 *E. faecalis* and 3 *E. faecium* considered weak producers. In our study at this temperature, A1.14.2022 (*E. faecalis*) and N9.25.2016 (*E. faecium*) exhibited moderate and weak biofilm production, respectively (Table 3).

Similarly, findings from Muruzović et al. [45], which investigated various incubation conditions including pH and NaCl alongside temperatures of 4, 20, and 37 °C, reinforced these conclusions. These authors observed that 37 °C not only supported planktonic cell growth but also facilitated biofilm formation. Additionally, they found that *E. hirae* exhibited lower biofilm production at 20 °C compared with 37 °C, and at 4 °C, none of the microorganisms in their study produced biofilm.

Collectively, these studies indicate that 37 °C was the optimal temperature for biofilm production by *Enterococcus* spp., while 4 °C could be limiting for both enterococcal biofilm formation and planktonic growth.

### 3.3. Biofilm Formation at Different NaCl Concentrations

In Portuguese traditional cheeses with PDO status, one of the three essential ingredients used in production—alongside milk and vegetable coagulant—is salt, which can reach concentrations of up to 25 g/L. For *Azeitão* cheeses, the NaCl concentration in the final product is 1%, whereas for *Nisa* cheeses, it is 2.5% [8]. To evaluate the biofilm-forming ability of the cheese enterococci under investigation, the isolates were exposed to a range of NaCl concentrations from 1% to 8%, in line with similar studies [71].

In terms of biofilm production classification for cheese enterococci under various NaCl conditions, it was noted that isolates A2.48.2016 and N10.27.2017 were non-producers across all evaluated concentrations and incubation periods (Table 4). While N10.27.2017 maintained its classification as a non-producer, A2.48.2016, initially categorized as a weak producer, was reclassified as a non-producer (Table 4, Figure 3). At 8% NaCl, most enterococci were classified as either weak, moderate, or even strong producers, with the exception of A4.16.2019 (Table 4, Figure 3). Notably, the highest biofilm production was observed at 48 h of incubation in 8% NaCl, in which two isolates (A1.6.2017 and A4.26.2021) were identified as strong producers (Figure 3). Additionally, A1.14.2022 consistently demonstrated increased biofilm production, being classified as either moderate or strong, across the various tested conditions (Table 4).

*Enterococcus* is known as a halotolerant bacterial genus, able to withstand and multiply in NaCl concentrations up to 6.5%. In terms of biofilm production, a study by Maurya et al. [71] reported that high NaCl concentrations, 8% after 24 h of incubation, had inhibitory effects on *E. faecium* recovered from sludge samples from an effluent treatment plant. These results are somewhat consistent with our study since the isolates N10.27.2017 and N9.25.2016, at 8% NaCl, were classified as non-producers and weak producers, respectively. Moreover, a study by Akoğlu [14] reported that the increase in NaCl caused a general decrease in biofilm formation for all the *Enterococcus* tested, except for *E. faecium*, which showed a slight increase in biofilm formation. Similarly to the results of Maurya et al. [71], these authors described a decrease in biofilm formation with the increase in NaCl concentration/percentage. Muruzović et al. [46] also investigated the effects of NaCl as a stressor on *E. hirae*, testing concentrations of 4, 6.5, and 8%, also reporting a decrease in biofilm production as NaCl concentration increased, a finding consistent with studies by Akoglu [14] and Maurya et al. [71]. Additionally, Solheim et al. [72] examined *E. faecalis* and the impact of NaCl-induced stress, noting that higher NaCl concentrations inhibited the expression of *gelE*, a gene critical for biofilm production in *E. faecalis* [72]. Overall, these results further highlight biofilm formation as a strain-specific characteristic, as evidenced by the discrepancies observed across distinct studies.

### 3.4. Biofilm Formation at Different pH Values

During the cheese manufacturing process, milk fermentation and subsequent biochemical reactions lead to various pH alterations [8]. *Enterococcus* spp. are renowned for their adaptability to diverse environmental conditions, capable of withstanding pH values ranging from 4 to 10. This remarkable tolerance allows them to persist throughout the cheesemaking process, from raw materials to the final product [73]. Consequently, for the present study, distinct pH conditions (5, 6, 7, 8, and 9) were selected to align closely with the pH levels of *Azeitão* (5.86) and *Nisa* (5.20) cheeses [6,8], as well as to incorporate conditions examined by other researchers, for comparative purposes [14,45,71,74].

The results obtained are summarized in Table 5 and Figure 4, highlighting a significant influence of pH on biofilm-forming abilities. A greater number of moderate and strong biofilm producers were observed compared with previous assays. As the pH increased, more enterococci were identified as biofilm producers; at pH 8 and 9, nearly all isolates were classified as such. Notably, after 48 h of incubation, an increased number of enterococci were categorized as weak, moderate, or strong producers (Figure 4, Table 5). However, the isolate N10.27.2017 consistently exhibited a non-producer phenotype across all tested conditions, while A1.14.2022 demonstrated significantly elevated biofilm production (classified as moderate or strong) compared with the other enterococci, with the exception of being classified as a weak producer at pH 5.

Maurya et al. [71] conducted assays to assess the biofilm formation ability of *E. faecium* at pH levels of 5, 6, 7, 7.5, and 8, reporting optimal production at pH 6, 7, and 9 when the incubation temperature was maintained at 37 °C. In our study, the two *E. faecium* isolates, N9.25.2016 and N10.27.2017, displayed either a non-producer profile or moderate production at pH 8 after 48 h, while exhibiting weak production at pH 9. This suggests a better adaptation to pH 8, aligning with the findings of Maurya et al. [71]. Additionally, they noted that lower pH levels led to decreased biofilm formation and planktonic cell growth.

Akoğlu [14] also explored pH levels of 5.5, 6.5, 7.5, and 8.5, revealing that the tested *E. faecalis* strain produced additional biofilm at the more acidic pH levels (5.5 and 6.5). Conversely, the *E. durans* strain exhibited low biofilm production across the various conditions, even at pH 7, which was used as the control [14]. In comparison, our study found that most *E. faecalis* isolates were classified as moderate producers in at least one of the acidic pH levels (Table 5). For *E. durans*, isolate A2.48.2016 was categorized as a non-producer, while A1.6.2017 was identified as a weak producer, corroborating the results of Akoğlu [14].

### 3.5. Biofilm Quantification Versus Cell Viability

Biofilm formation is known to be triggered by various stressors, in order to assure microbial survival and subsequent proliferation in a given environment. This survival strategy relies not only on the biofilm EPS matrix produced but also on cell viability, either to ensure further proliferation after stress exposer or even to produce the biofilm itself [64].

The association between biofilm production and cell viability, throughout the different assays, showed a variety of distinct behaviors among the enterococci under study. Some cheese isolates showed high biofilm production, associated with low cell viability (N9.1.2019, pH 7 at 24 h, for instance), or low biofilm production and high viability (e.g., A4.16.2019, pH 5, 6 at 24 h) (Appendix A). Therefore, to further explore this subject, a principal component analysis (PCA) was performed.

PCA allowed the establishment of correlations between all variables tested (details in Appendix A) by deployment in different dimensions (Appendix A). Three principal components were used to explain the data variability (PC1, Dim-1; PC2, Dim-2; and PC3, Dim-3), which together accounted for 82.3% of the data variability. Dim-1 was able to explain biofilm production in all assays and almost all cell viability results, except for pH 9 at 48 h (explained by Dim-2), and NaCl at 8% at 24–48 h and pH 8 at 48 h (explained by Dim-3).

Regarding the spatial representation of PC1/PC2/PC3, any isolate projected closer to the positive end of Dim-1 would be expected to have higher values in the explanatory variables of PC1 and reduced values in PC2 and PC3 and vice versa (e.g., A1.14.2022 and N10.27.2017). Hence, the projection of the microorganism in the diagram, especially in Dim-1, was dependent on biofilm production ability but also on cell viability since Dim-1 accounted for both measurements (with few exceptions). In this case it was possible to observe two extremes: A1.14.2022, a strong biofilm producer in most conditions tested, and N10.27.2017, with non-producer profiles in almost all conditions (Appendix A). In the case of the variables explained by PC3, it was possible to see isolate N10.55.2022, classified as moderate biofilm producer at pH 8 and strong at pH 9. As for Dim-3, it displayed isolates A1.6.2017 and A4.26.2021, classified as strong biofilm producers in NaCl 8% at 24 and 48 h.

Overall, PCA indicated a proportionality between biofilm production and cell viability for the enterococci and conditions under study. The justifying factor for this assumption was related to the positive gradient of Dim-1 as it explained the variables of quantification of produced biofilm and cell viability simultaneously. Thus, and as already mentioned, the microorganisms were distributed on this axis according to their biofilm production ability and corresponding cell viability, being expected that the more biofilm was produced, the greater the viability of the cells in the biofilm.

To further verify this assumption, calculations were performed to determine biofilm absorbance units per viable cell (Figure 5, Figure 6 and Figure 7 and Appendix A). The results are presented for each isolate condition, with an extra line added at the end of each graphic representation to depict the average behavior of all cheese enterococci under the tested conditions.

This innovative analytical approach not only allowed us to identify the condition associated with higher rates of biofilm production per viable cell for each specific condition—temperature, NaCl concentration, or pH values (e.g., in Figure 6, A1.6.2017, 37 °C at 24 h, or in Figure 8, N10.27.2017, pH 6 at 48 h) and isolate (e.g., for A4.16.2019, 8% NaCl at 24 h, or for N9.1.2019, pH 7 at 24 h), but also highlighted the conditions that resulted in the highest rates (green bubbles in the graphs) (for temperature, 37 °C at 24/48 h; for NaCl concentration, 8% at 24/48 h; and for pH values, pH 9 at 24 h and pH 6 at 48 h).

Notably, isolate A1.14.2022 consistently displayed values above the average in most conditions, reinforcing its status as the top biofilm producer across all tested incubation conditions. In contrast, N10.27.2017 exhibited values below the average in most cases and showed lower levels compared with other isolates in this study (Appendix A).

To further confirm which cheese enterococci exhibit higher, or lower, biofilm formation ability under the studied conditions, all obtained data were used to construct a similarity dendrogram (Figure 8).

Overall, the dendrogram clearly distinguished isolate A1.14.2022 in cluster I as the best biofilm producer across all tested conditions. In contrast, N10.27.2017 was distinctly separated in cluster III as the poorest biofilm producer throughout the assays. Cluster II comprised isolates that generally exhibited intermediate biofilm production capabilities. This cluster organization aligned with the PCA, which also highlighted A1.14.2022 as the isolate most proficient in biofilm production, also demonstrating higher cell viability in all assays (Appendix A). Cluster II, as noted, aggregated isolates with moderate biofilm production and lower cell viability, with N9.1.2019 and N10.55.2022 generally exhibiting higher cell viability percentages compared with other isolates within the same cluster (Appendix A). As for cluster III, a distinct separation was observed between N10.27.2017, identified as having the lowest biofilm production and cell viability, and A2.48.2016 and N9.25.2016, which, while exhibiting low biofilm production, demonstrated some degree of cell viability in certain cases (Appendix A).

### 3.6. The Role of Biofilm Formation in Traditional Cheese Making

The significance of biofilms in the food industry has evolved over the years. When composed of pathogenic and/or spoilage microorganisms, these structures are regarded as one of the most pressing challenges in food processing [49], being associated with cross-contamination and apparatus malfunctioning [34,49,75]. In fact, biofilms are recognized as a primary source of food contamination, arising from direct contact with surfaces and equipment in the processing line that are conducive to biofilm formation. This issue is particularly concerning for microorganisms such as *Listeria monocytogenes* and numerous other foodborne pathogens [31,38,39,76]. However, biofilms are increasingly being regarded in a more positive light due to their beneficial effects in areas such as wastewater treatment and their application in food products, including edible biofilms [40].

In fact, due to being one of the main sources of contamination in food and a resistance/persistence structure, biofilms are believed to play a role in maintaining the autochthonous microbiota of traditional cheeses. Therefore, the production of biofilms by the autochthonous microbiota (LAB) itself—on the surfaces of the production environment, the equipment, and within the cheese matrix—enhances the persistence and maintenance of this microbiota in the original cheese dairies. Practical cases observed in the Portuguese cheese industry support this belief, as further explained below.

In the first case, the replacement of equipment in the dairies had a significant impact on the cheeses during the maturation period, which were previously deposited on wooden shelves. However, due to increased sanitation requirements by the European Commission and as established by HACCP (Sanitation Plan) systems, these wooden shelves have been replaced with aluminum ones [77,78]. In the first batches produced after this substitution, changes in the organoleptic properties of the traditional cheeses were observed. This small change in material had significant impacts as the taste and aroma of the cheese were completely altered, even though the entire manufacturing process remained consistent over time.

Wood is a more porous material and thus more prone to the development of microorganism aggregates. Additionally, the arrangement of the cheese during maturation was another determining factor. The newer cheeses were placed at the top of the shelf, moving the previous batch to the back row, so that the older cheeses were at the bottom of the shelves. This allowed the microorganisms from previous batches, housed in the wood, to come into contact with the newer cheeses, facilitating their colonization by the native microbiota of the original cheese factory (Personal communication n.d.).

The second case was reported to the authors by the engineer responsible for production at one of the cheese dairies producing Portuguese PDO cheeses in this study. According to the engineer, the cheese factory regularly cleans its equipment using more abrasive cleaning and disinfection products. Under normal conditions, during their maturation period, the cheeses pass through several maturation chambers, where they remain for a defined period (days). In the first maturation chamber, where the cheeses stay for a specific and predetermined period, the native microbiota of the cheese (non-starter cultures) begins its active development. It was found that after these more thorough/aggressive cleanings in the cheese factory, the first produced batches of cheeses could not achieve the desired characteristics during the maturation time. Consequently, to obtain the expected organoleptic characteristics associated with the product, it became necessary to extend the maturation period in the first chamber (Personal communication n.d.).

Even though the experimental medium used in this study (TSB supplemented with 1.5% glucose) did not mimic the complexity of milk, cheese matrixes, manufacturing and maturing surfaces used in cheesemaking factories, or the autochthonous microbial communities already present, this study undoubtedly shows the potentiality of these cheese enterococci to produce biofilm and their positive influence on the maintenance of factory-specific cheese microbiota. Moreover, the cases highlighted above emphasize the critical role of microbial biofilms in sustaining the autochthonous microbiota within cheese dairies. They also underscore the necessity for further research to validate these findings and to explore the technological potential of these microorganisms.

## 4. Conclusions

Enterococci are ubiquitous in all Portuguese traditional cheeses with PDO status, recognized for their beneficial roles in cheese production, primarily through proteolytic activity and the generation of volatile compounds that enhance flavor and aroma. Given that the production of these cheeses does not allow the addition of starter or non-starter lactic acid bacteria, their distinctive characteristics stem solely from the microbiota present in the milk and the specific cheese factory of origin.

This paper suggests that the variations among traditional cheeses may be linked to the persistence of cheese factory-specific microbiota. These microorganisms not only facilitate the cheesemaking process but also preserve the unique attributes characteristic of each cheese factory, thus differentiating them within and across regions. In the case of *Enterococcus* spp. isolates from *Azeitão* and *Nisa*, all demonstrated the ability to produce biofilm, although to varying degrees. This biofilm production enables these microorganisms to establish themselves across different cheese factories and to colonize various batches of cheese produced over the years, thereby sustaining factory-specific features and fostering a microbial heritage passed down through generations.

A comparative analysis of biofilm production and associated cell viability yielded interesting results. Utilizing an integrated approach based on PCA, a general correlation was observed between biofilm formation and cell viability.

Furthermore, the study indicated that as the pH increased, more enterococci were identified as biofilm producers. At pH 8 and 9, nearly all isolates were classified as such, suggesting that pH represented the most challenging condition during cheese manufacture. Nevertheless, the studied enterococci appeared well adapted to endure these fluctuations.

Overall, these findings highlight the significance of understanding the role of enterococci and their biofilm-forming capabilities in preserving the unique characteristics of traditional Portuguese cheeses, ultimately contributing to their quality and heritage.

## Figures and Tables

**Figure 1 foods-13-03067-f001:**
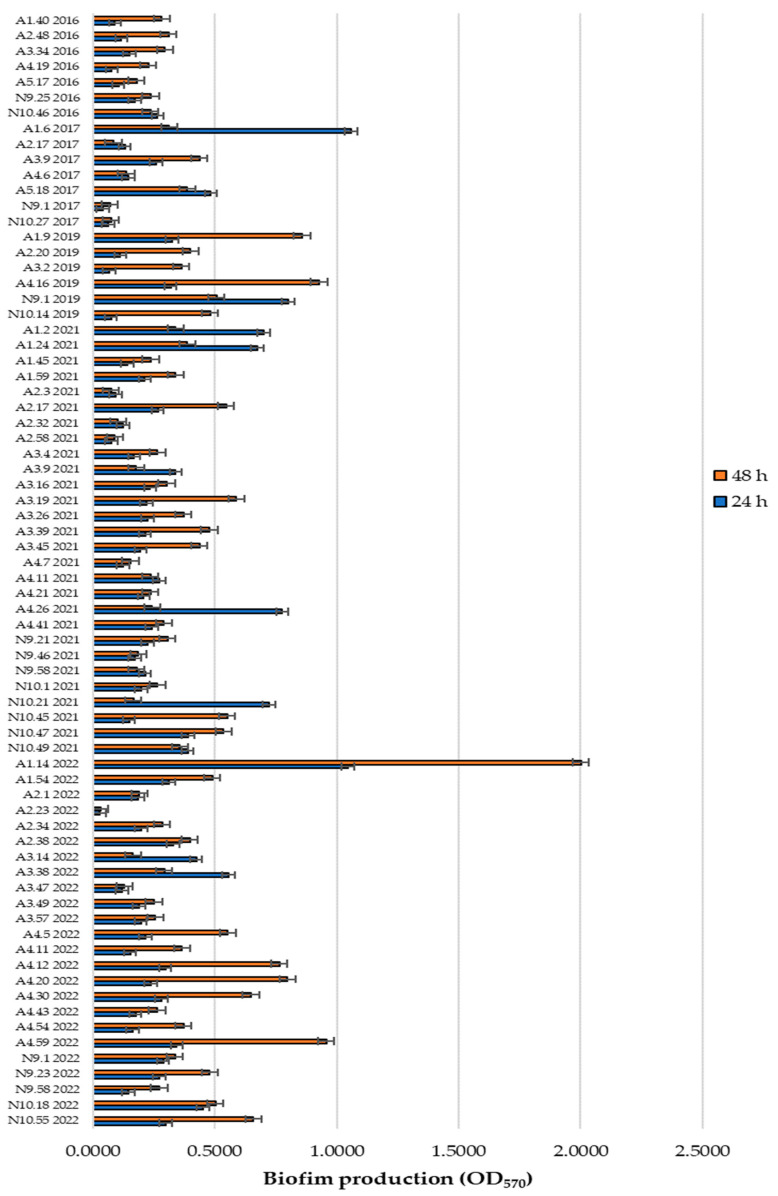
Biofilm production in TSB + 1.5% glucose, under incubation periods of 24 or 48 h. Legend: A—Azeitão; N—Nisa. A1–A5—*Azeitão* cheese factories; N9–N10—*Nisa* cheese factories.

**Figure 2 foods-13-03067-f002:**
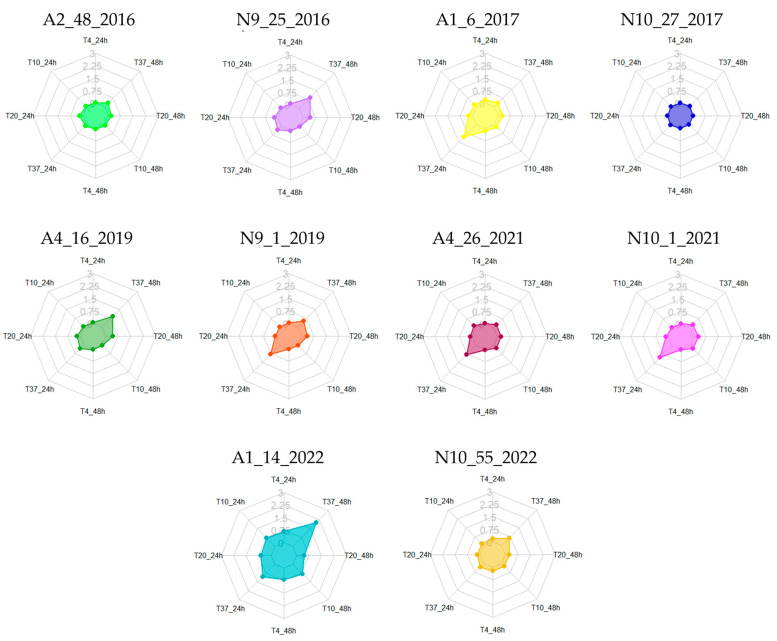
Biofilm production at different temperatures (4, 10, 20, or 37 °C) and incubation periods (24 or 48 h). Legend: A1–A5—*Azeitão* cheese factories; N9–N10—*Nisa* cheese factories. The scale used for the graphs represents biofilm quantification after crystal violet staining (OD_570_) nm.

**Figure 3 foods-13-03067-f003:**
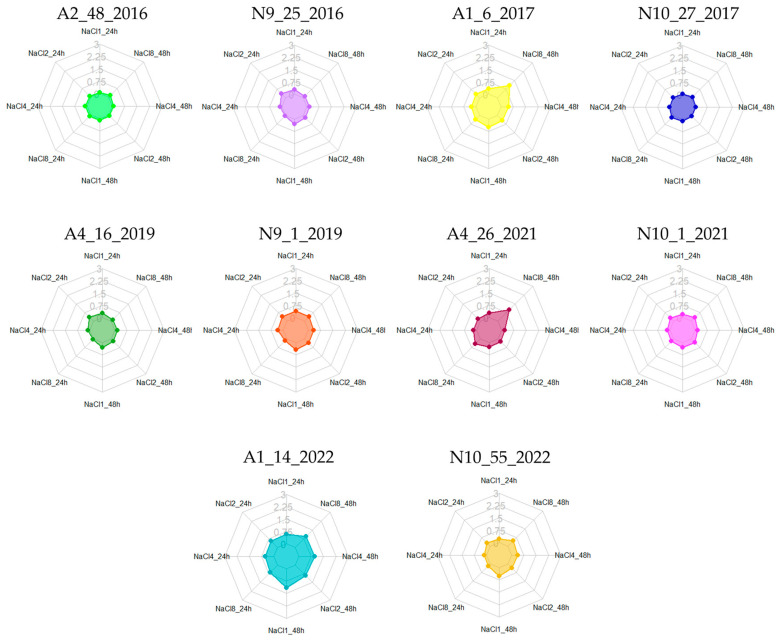
Biofilm production at different NaCl concentrations (1, 2, 4, and 8%) and incubation periods (24 and 48 h). Legend: A1–A5—*Azeitão* cheese factories; N9–N10—*Nisa* cheese factories. The scale used for the graphs represents biofilm quantification after crystal violet staining (OD_570_) nm.

**Figure 4 foods-13-03067-f004:**
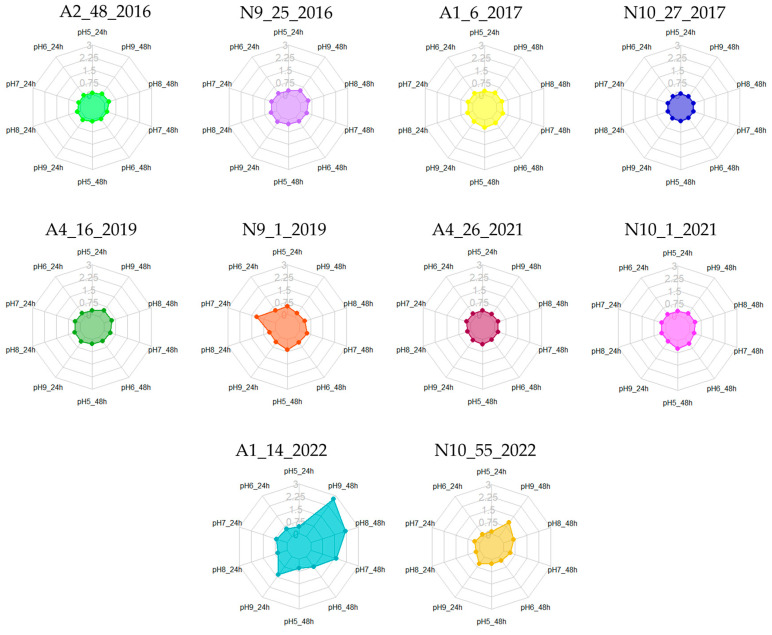
Biofilm production at different pH values (5, 6, and 7) and incubation periods (24 and 48 h). Legend: A1–A5—*Azeitão* cheese factories; N9–N10—*Nisa* cheese factories. The scale used for the graphs represents biofilm quantification after crystal violet staining (OD_570_) nm.

**Figure 5 foods-13-03067-f005:**
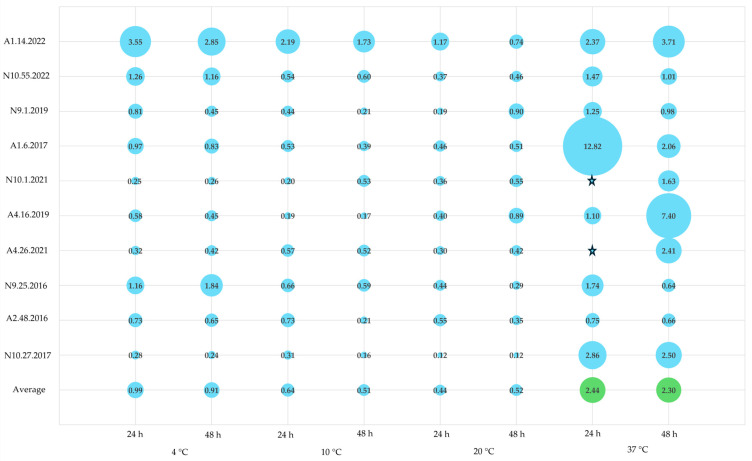
Biofilm production per viable cells at different temperatures at both incubation periods (24 or 48 h), as well as average production. (*) Identifies isolates with no detectable viability. Bubbles with green color identify the conditions with higher production of biofilm (on average).

**Figure 6 foods-13-03067-f006:**
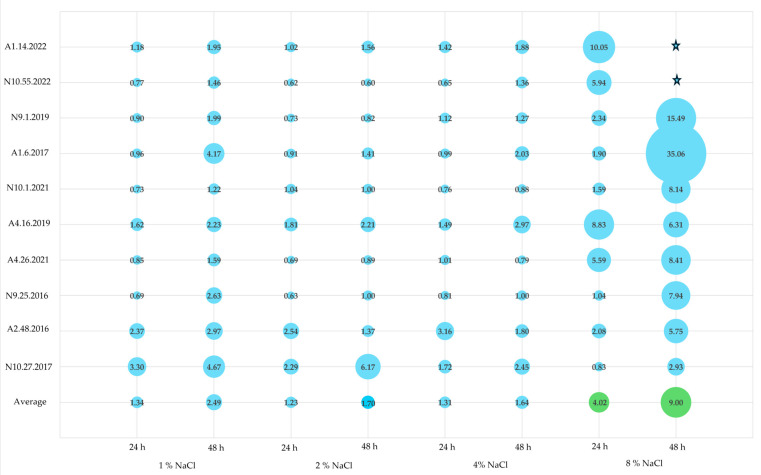
Biofilm production per viable cells at different NaCl concentrations at both incubation periods (24 or 48 h), as well as average production. (*) Identifies isolates with no detectable viability. Bubbles with green color identify the conditions with higher production of biofilm (on average).

**Figure 7 foods-13-03067-f007:**
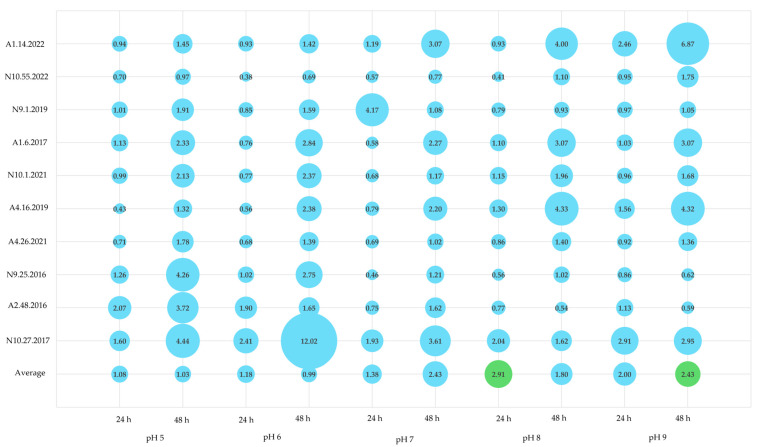
Biofilm production per viable cells at different pH values at both incubation periods (24 or 48 h), as well as average production. Bubbles with green color identify the conditions with higher production of biofilm (on average).

**Figure 8 foods-13-03067-f008:**
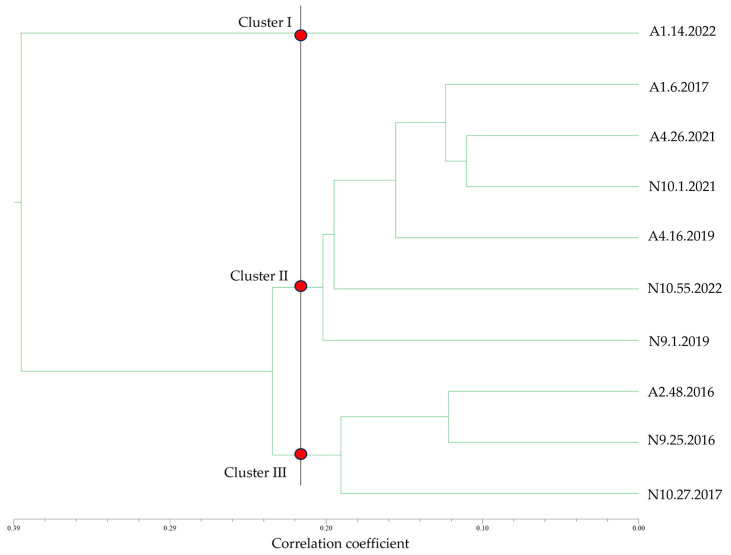
Similarity dendrogram, constructed using the Euclidean distance as a measure of association and the agglomeration method based on the unweighted pair group method with arithmetic mean (UPGMA). Legend: A—Azeitão cheese factories; N—Nisa cheese factories. The black line delineates the boundaries between the different clusters of biofilm producers. Cluster I—best producer; cluster II—intermediate producers; cluster III—worst producers.

**Table 1 foods-13-03067-t001:** Classification of biofilm production, for the enterococci under study, in two incubation periods at 37 °C.

Isolates	24 h	48 h	Isolates	24 h	48 h	Isolates	24 h	48 h	Isolates	24 h	48 h
A1.40 2016	N.P.	W.P.	N9.1 2019	M.P.	M.P.	A4.11 2021	W.P.	N.P.	A3.14 2022	W.P.	N.P.
A2.48 2016	N.P.	W.P.	N10.14 2019	N.P.	M.P.	A4.21 2021	N.P.	N.P.	A3.38 2022	M.P.	W.P.
A3.34 2016	N.P.	W.P.	A1.2 2021	M.P.	W.P.	A4.26 2021	M.P.	W.P.	A3.47 2022	N.P.	N.P.
A4.19 2016	N.P.	N.P.	A1.24 2021	M.P.	W.P.	A4.41 2021	W.P.	W.P.	A3.49 2022	N.P.	W.P.
A5.17 2016	N.P.	N.P.	A1.45 2021	N.P.	W.P.	N9.21 2021	N.P.	W.P.	A3.57 2022	N.P.	W.P.
N9.25 2016	N.P.	W.P.	A1.59 2021	N.P.	W.P.	N9.46 2021	N.P.	N.P.	A4.5 2022	N.P.	M.P.
N10.46 2016	W.P.	N.P.	A2.3 2021	N.P.	N.P.	N9.58 2021	N.P.	N.P.	A4.11 2022	N.P.	W.P.
A1.6 2017	S.P.	W.P.	A2.17 2021	W.P.	M.P.	N10.1 2021	N.P.	W.P.	A4.12 2022	W.P.	M.P.
A2.17 2017	N.P.	N.P.	A2.32 2021	N.P.	N.P.	N10.21 2021	M.P.	N.P.	A4.20 2022	W.P.	M.P.
A3.9 2017	W.P.	W.P.	A2.58 2021	N.P.	N.P.	N10.45 2021	N.P.	M.P.	A4.30 2022	W.P.	M.P.
A4.6 2017	N.P.	N.P.	A3.4 2021	N.P.	W.P.	N10.47 2021	W.P.	M.P.	A4.43 2022	N.P.	W.P.
A5.18 2017	M.P.	W.P.	A3.9 2021	W.P.	N.P.	N10.49 2021	W.P.	W.P.	A4.54 2022	N.P.	W.P.
N9.1 2017	N.P.	N.P.	A3.16 2021	N.P.	W.P.	A1.14 2022	S.P.	S.P.	A4.59 2022	W.P.	W.P.
N10.27 2017	N.P.	N.P.	A3.19 2021	N.P.	M.P.	A1.54 2022	W.P.	M.P.	N9.1 2022	W.P.	W.P.
A1.9 2019	W.P.	M.P.	A3.26 2021	N.P.	W.P.	A2.1 2022	N.P.	N.P.	N9.23 2022	W.P.	M.P.
A2.20 2019	N.P.	W.P.	A3.39 2021	N.P.	M.P.	A2.23 2022	N.P.	N.P.	N9.58 2022	N.P.	W.P.
A3.2 2019	N.P.	W.P.	A3.45 2021	N.P.	W.P.	A2.34 2022	N.P.	W.P.	N10.18 2022	W.P.	M.P.
A4.16 2019	W.P.	M.P.	A4.7 2021	N.P.	N.P.	A2.38 2022	W.P.	W.P.	N10.55 2022	W.P.	M.P.

Legend: The color scale indicates the non-producers or weak producers with lighter shades of green and the moderate and strong producers with darker shades. N.P—non-producer; W.P.—weak producer; M.P.—moderate producer; S.P.—strong producer (according to the thresholds explained in the material and methods section).

**Table 2 foods-13-03067-t002:** Classification of biofilm production for the selected isolates, after 24 or 48 h at 37 °C.

	48 h	N.P.	W.P.	M.P.	S.P.
24 h					
N.P.		N9.25.2016 N10.27.2017	A2.48.2016 N10.1.2021		
W.P.				A4.16.2019 N10.55.2022	
M.P.			A4.26.2021	N9.1.2019	
S.P.			A1.6.2017		A1.14.2022

**Table 3 foods-13-03067-t003:** Classification of biofilm production at different temperatures (4, 10, 20, or 37 °C) and incubation periods (24 or 48 h).

Temperature								
	4 °C		10 °C		20 °C		37 °C	
Isolates	24 h	48 h	24 h	48 h	24 h	48 h	24 h	48 h
A2.48.2016	N.P.	N.P.	N.P.	N.P.	N.P.	N.P.	N.P.	W.P.
N9.25.2016	N.P.	W.P.	N.P.	N.P.	N.P.	N.P.	N.P.	N.P.
A1.6.2017	N.P.	N.P.	N.P.	N.P.	W.P.	W.P.	W.P.	W.P.
N10.27.2017	N.P.	N.P.	N.P.	N.P.	N.P.	N.P.	N.P.	N.P.
A4.16.2019	N.P.	N.P.	N.P.	N.P.	N.P.	W.P.	W.P.	M.P.
N9.1.2019	N.P.	N.P.	N.P.	N.P.	N.P.	W.P.	M.P.	M.P.
A4.26.2021	N.P.	N.P.	N.P.	N.P.	N.P.	N.P.	M.P.	W.P.
N10.1.2021	N.P.	N.P.	N.P.	W.P.	N.P.	W.P.	N.P.	W.P.
A1.14.2022	M.P.	M.P.	M.P.	M.P.	M.P.	W.P.	S.P.	S.P.
N10.55.2022	N.P.	N.P.	N.P.	N.P.	N.P.	W.P.	W.P.	M.P.

Legend: The color scale indicates the non-producers or weak producers with lighter shades of green and the moderate and strong producers with darker shades. N.P—non-producer; W.P.—weak producer; M.P.—moderate producer; S.P.—strong producer.

**Table 4 foods-13-03067-t004:** Classification of biofilm production at different NaCl concentrations (1, 2, 4, and 8%) and incubation periods (24 and 48 h).

NaCl Concentration								
	1%		2%		4%		8%	
Isolates	24 h	48 h	24 h	48 h	24 h	48 h	24 h	48 h
A2.48.2016	N.P.	N.P.	N.P.	N.P.	N.P.	N.P.	N.P.	N.P.
N9.25.2016	N.P.	N.P.	N.P.	N.P.	N.P.	N.P.	W.P.	W.P.
A1.6.2017	W.P.	M.P.	W.P.	W.P.	W.P.	W.P.	W.P.	S.P.
N10.27.2017	N.P.	N.P.	N.P.	N.P.	N.P.	N.P.	N.P.	N.P.
A4.16.2019	W.P.	W.P.	W.P.	N.P.	N.P.	N.P.	N.P.	N.P.
N9.1.2019	W.P.	W.P.	W.P.	W.P.	W.P.	W.P.	N.P.	W.P.
A4.26.2021	W.P.	W.P.	N.P.	N.P.	N.P.	N.P.	W.P.	S.P.
N10.1.2021	N.P.	W.P.	W.P.	W.P.	N.P.	N.P.	N.P.	W.P.
A1.14.2022	M.P.	S.P.	M.P.	M.P.	M.P.	S.P.	M.P.	M.P.
N10.55.2022	N.P.	M.P.	W.P.	W.P.	N.P.	W.P.	N.P.	M.P.

Legend: The color scale indicates the non-producers or weak producers with lighter shades of green and the moderate and strong producers with darker shades. N.P—non-producer; W.P.—weak producer; M.P.—moderate producer; S.P.—strong producer.

**Table 5 foods-13-03067-t005:** Classification of biofilm production at different pH values (5, 6, 7, 8, and 9) and incubation periods (24 and 48 h).

pH Values										
	pH 5		pH 6		pH 7		pH 8		pH 9	
Isolates	24 h	48 h	24 h	48 h	24 h	48 h	24 h	48 h	24 h	48 h
A2.48.2016	N.P.	N.P.	N.P.	N.P.	N.P.	N.P.	N.P.	W.P.	N.P.	N.P.
N9.25.2016	N.P.	N.P.	N.P.	N.P.	N.P.	N.P.	N.P.	M.P.	N.P.	W.P.
A1.6.2017	W.P.	W.P.	W.P.	W.P.	W.P.	W.P.	W.P.	W.P.	W.P.	W.P.
N10.27.2017	N.P.	N.P.	N.P.	N.P.	N.P.	N.P.	N.P.	N.P.	N.P.	N.P.
A4.16.2019	N.P.	W.P.	W.P.	W.P.	W.P.	W.P.	W.P.	M.P.	W.P.	M.P.
N9.1.2019	M.P.	M.P.	W.P.	W.P.	S.P.	M.P.	W.P.	W.P.	W.P.	W.P.
A4.26.2021	N.P.	W.P.	N.P.	N.P.	W.P.	N.P.	N.P.	W.P.	W.P.	N.P.
N10.1.2021	W.P.	M.P.	W.P.	W.P.	W.P.	W.P.	W.P.	W.P.	W.P.	W.P.
A1.14.2022	W.P.	M.P.	M.P.	M.P.	M.P.	S.P.	M.P.	S.P.	S.P.	S.P.
N10.55.2022	N.P.	W.P.	N.P.	W.P.	W.P.	W.P.	N.P.	M.P.	M.P.	S.P.

Legend: The color scale indicates the non-producers or weak producers with lighter shades of green and the moderate and strong producers with darker shades. N.P—non-producer; W.P.—weak producer; M.P.—moderate producer; S.P.—strong producer.

## Data Availability

The original contributions presented in the study are included in the article and Appendix A, further inquiries can be directed to the corresponding author.

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
