# Peer review of "Beyond Harmful: Exploring Biofilm Formation by Enterococci Isolated from Portuguese Traditional Cheeses"

_foods, 2024, doi:10.3390/foods13193067_

Round 1

Reviewer 1 Report

Comments and Suggestions for Authors

The manuscript submitted by Serrano S. et al., titled "Beyond Harmful: Exploring Biofilm Formation by Enterococci Isolated from Portuguese Traditional Cheeses " provides valuable insights into the presence of Enterococcus in Portuguese artisanal cheeses and its biofilm production. However, I recommend revising the tables and figures to enhance the manuscript's clarity and impact.

In particular, I suggest the following changes:

Tables 1, 2, and 3: These tables should either be removed or included as supplementary material.

Table 1 detail the isolates, are described in reference 23? (Rocha et al., 2022).

Table 2 describes conditions that could be more effectively integrated into the text.

Table 3 could be removed, with relevant information incorporated into the Materials and Methods section and the legend for Table 4 in the original manuscript.

Figure 1: Consider reducing the figure to show only selected strains (16 strains), and rotate it 90 degrees for better visualization.

Table 5: This table appears unnecessary and could be omitted.

Figure 5: I suggest moving this figure to the supplementary materials.

These modifications will likely improve the manuscript's readability and make the data and results more accessible to readers.

Author Response

Reviewer 1

The manuscript submitted by Serrano S. et al., titled "Beyond Harmful: Exploring Biofilm Formation by Enterococci Isolated from Portuguese Traditional Cheeses " provides valuable insights into the presence of Enterococcus in Portuguese artisanal cheeses and its biofilm production. However, I recommend revising the tables and figures to enhance the manuscript's clarity and impact.

In particular, I suggest the following changes:

The authors thank the reviewer for the suggestions which we took into consideration and hopefully ameliorated the manuscript.

Tables 1, 2, and 3: These tables should either be removed or included as supplementary material.

Answer: Table 1 was included as supplementary material and the remaining tables deleted, adding their information to the text.

Table 1 detail the isolates, are described in reference 23? (Rocha et al., 2022).

Answer: The reference of Rocha et al., (2022) describes the isolation methodology used to acquire the cheese enterococci collection, however the collection used in the paper Rocha et al., (2022) lacks isolates from the years of 2021 and 2022 depicted in the present manuscript.

Table 2 describes conditions that could be more effectively integrated into the text.

Answer: The table’s information was added to the text (lines 159-160) and the table deleted.

Table 3 could be removed, with relevant information incorporated into the Materials and Methods section and the legend for Table 4 in the original manuscript.

Answer: The table’s information was added to the text (lines 198-200) and table deleted.

Figure 1: Consider reducing the figure to show only selected strains (16 strains) and rotate it 90 degrees for better visualization.

Answer: Figure 1 depicts all the isolates selected, from a larger collection, for the biofilm production assay. Hence, the authors believe that its reduction to only the selected strains would mislead the reader into thinking that the collection was composed only of the selected isolates. Consequently, the authors consider that for a better understanding of the magnitude of the enterococci collection that Figure 1 should remain as such.

Table 5: This table appears unnecessary and could be omitted.

Answer: Table 5 (now Table 2) provides a summary of the selected isolates for further testing, along with their classification as biofilm producers. This table enables the reader to rapidly identify both the selected isolates and their capacity to produce biofilm at both 24 and 48 hours, facilitating comparison between the different incubation conditions, throughout the research paper, without having to search for that information in the larger table, where all 72 isolates are included (Table 1). Accordingly, the authors considered that it should remain in the manuscript.

Figure 5: I suggest moving this figure to the supplementary materials.

Answer: This figure has been moved to the supplementary materials.

These modifications will likely improve the manuscript's readability and make the data and results more accessible to readers.

Reviewer 2 Report

Comments and Suggestions for Authors

This study investigated the biofilm-forming capabilities of Enterococcus isolates from Portuguese traditional cheeses and highlighted their critical roles in the unique characteristics of traditional cheeses. The manuscript is well organized and nicely written. The work of this manuscript is so systematic and enormous, the discussion is sufficient. However, I have minor points as follows:

1. Line 97-98, appropriate references should been inserted at the end of this sentence, rather than at the end of this paragraph.

2. Line 142-145, at maximum speed at specific values should be given.

3. Line 237-238, why the biofilm formation capacity was observed to decrease after 48 h incubation compared with 24 h? Please explain the possible reason for this.

4. The authors used TSB as a basal medium to examine the biofilm forming ability of all strains. However, the nutrient environment that Enterococcus encounter in cheese production differs significantly from this medium. Therefore, it is important for the authors to examine whether the experimental results obtained in this study accurately reflect the real conditions. Discussion or further investigation into this matter is warranted.

Author Response

Reviewer 2

This study investigated the biofilm-forming capabilities of Enterococcus isolates from Portuguese traditional cheeses and highlighted their critical roles in the unique characteristics of traditional cheeses. The manuscript is well organized and nicely written. The work of this manuscript is so systematic and enormous, the discussion is sufficient. However, I have minor points as follows:

The authors thank the reviewer for the suggestions which we took into consideration and hopefully ameliorated the manuscript.

  1. Line 97-98, appropriate references should been inserted at the end of this sentence, rather than at the end of this paragraph.

Answer: We have added the references corresponding to that sentence (see line 99).

  1. Line 142-145, at maximum speed at specific values should be given.

Answer: The specific values were added to the manuscript (see lines 145 and 148).

  1. Line 237-238, why the biofilm formation capacity was observed to decrease after 48 h incubation compared with 24 h? Please explain the possible reason for this.

Answer: the text was slightly changed to be clearer and an explanation of the behaviour of same strains, namely strain A1.6.2017, is already in the 259-265.

  1. The authors used TSB as a basal medium to examine the biofilm forming ability of all strains. However, the nutrient environment that Enterococcus encounter in cheese production differs significantly from this medium. Therefore, it is important for the authors to examine whether the experimental results obtained in this study accurately reflect the real conditions. Discussion or further investigation into this matter is warranted.

Answer: This is a highly pertinent and insightful question. The authors are aware that the medium used for the assays does not resemble the ones encountered in cheese factories or cheese-making environments. This is because it is known that both milk and cheese matrices are very complex and rich media. However, the use of milk as a growth medium presents several challenges in experimental assays designed to assess the biofilm production ability of isolates derived from dairy products. These challenges include the potential for milk spoilage under the various conditions tested throughout this research paper. Furthermore, the fat content of milk could potentially interfere with the dyes used for biofilm quantification using crystal violet and cell viability using resazurine. This could lead to inaccurate results and hinder any conclusions regarding the biofilm production capability of cheese enterococci. As this methodology has not been previously described, the results would not be comparable to other peer-reviewed works. Furthermore, another challenge in creating a realistic cheese factory environment would be to replicate the microbial communities present at the various stages of Portuguese traditional PDO cheese production. The microbial communities present in the cheese factories under study, along with their interactions, have yet to be investigated throughout the various stages of cheese manufacturing. Therefore, it is exceedingly challenging to accurately simulate these environments, and this is not the objective of the present study. The authors' primary aim was to demonstrate that these cheese enterococci are capable of producing biofilms and that, due to the manufacturing practices employed for this type of cheese, biofilm production is a beneficial trait that allows the maintenance of microbial heritage over time, thereby ensuring the continued high quality of Portuguese PDO cheeses.

The authors have added discussion addressing this matter in section “3.6. The Role of Biofilm Formation in Traditional Cheese Making” as suggested and appreciate the thoughtful and interesting question (see lines 573-578).

Round 2

Reviewer 1 Report

Comments and Suggestions for Authors

the present version of the ms is accepted for publication